# What Is the Significance of Indeterminate Pulmonary Nodules in High-Grade Soft Tissue Sarcomas? A Retrospective Cohort Study

**DOI:** 10.3390/cancers15133531

**Published:** 2023-07-07

**Authors:** Marcus J. Brookes, Corey D. Chan, Timothy P. Crowley, Maniram Ragbir, Thomas Beckingsale, Kanishka M. Ghosh, Kenneth S. Rankin

**Affiliations:** 1North of England Bone and Soft Tissue Tumour Service, Royal Victoria Infirmary, Queen Victoria Road, Newcastle upon Tyne NE1 4LP, UK; corey.chan@newcastle.ac.uk (C.D.C.); timothy.crowley1@nhs.net (T.P.C.); maniram.ragbir@nhs.net (M.R.); t.beckingsale@nhs.net (T.B.); kanishka.ghosh1@nhs.net (K.M.G.); kenneth.rankin@newcastle.ac.uk (K.S.R.); 2Translational and Clinical Research Institute, Newcastle University, Newcastle upon Tyne NE1 7RU, UK

**Keywords:** indeterminate pulmonary nodules (IPNs), sarcoma, soft tissue sarcoma, survival, metastasis, metastases

## Abstract

**Simple Summary:**

Sarcomas are rare cancers; they can arise anywhere in the body and most often spread to the lungs. When patients are diagnosed, they have a scan of the chest to look for this. The scan often finds small nodules whereby we cannot be certain whether they are cancer or not; these are called indeterminate pulmonary nodules or IPNs. We do not yet understand what the presence of IPNs means for patients with high-grade sarcomas in their soft tissues, although we know that some of these reveal themselves later on as being a spreading of the cancer. Currently, patients with IPNs normally have repeat scans a number of months down the line to see whether they have changed in size, suggesting that they may be cancer. This study has identified a number of different characteristics that make these IPNs more likely to be cancer.

**Abstract:**

Background: Sarcomas are rare, aggressive cancers which frequently metastasise to the lungs. Following diagnosis, patients typically undergo staging by means of a CT scan of their chest. This often identifies indeterminate pulmonary nodules (IPNs), but the significance of these in high-grade soft tissue sarcoma (STS) is unclear. Identifying whether these are benign or malignant is important for clinical decision making. This study analyses the clinical relevance of IPNs in high-grade STS. Methods: All patients treated at our centre for high-grade soft tissue sarcoma between 2010 and 2020 were identified from a prospective database. CT scans and their reports were reviewed, and survival data were collected from patient records. Results: 389 suitable patients were identified; 34.4% had IPNs on their CT staging scan and 20.1% progressed into lung metastases. Progression was more likely with IPNs ≥ 5 mm in diameter (*p* = 0.006), multiple IPNs (*p* = 0.013) or bilateral IPNs (*p* = 0.022), as well as in patients with primaries ≥ 5 cm (*p* = 0.014), grade 3 primaries (*p* = 0.009) or primaries arising deep to the fascia (*p* = 0.041). The median time to progression was 143 days. IPNs at diagnosis were associated with an increased risk of developing lung metastases and decreased OS in patients with grade 3 STS (*p* = 0.0019 and *p* = 0.0016, respectively); this was not observed in grade 2 patients. Conclusions: IPNs at diagnosis are associated with significantly worse OS in patients with grade 3 STS. It is crucial to consider the primary tumour as well as the IPNs when considering the risk of progression. Surveillance CT scans should be carried out within 6 months.

## 1. Introduction

Sarcomas are rare [1], aggressive tumours arising from mesenchymal cells which most commonly metastasise to the lungs [2]; up to 30% of soft tissue sarcoma (STS) patients present with synchronous metastases [3,4,5,6]. Following diagnosis, guidelines dictate that patients should undergo staging by means of a CT scan of their chest [7,8]. Whilst CT scanning has good utility for the identification of metastases, allowing important treatment decisions to be made, they often identify indeterminate pulmonary nodules (IPNs), but the significance of these is currently unclear.

The prevalence of IPNs varies widely in the literature; Rissing et al. reported IPNs at diagnosis in 21% of the 331 sarcoma patients whom they followed prospectively [5], whilst a retrospective review by Saifuddin et al. identified IPNs in up to 49.5% of 200 STS patients [6]. Whilst there is variation in CT scanning modalities, differences in technology and variation in radiological reporting that may explain some of this; there remains uncertainty surrounding IPNs. There is no agreed definition of what an IPN is and what metastatic disease is based on CT imaging, and as such, there is a high degree of variability; non-calcified nodules either <5 mm or <10 mm in size are frequently used [3,5,6]. Arguably more importantly, with regard to the significance of IPNs in STS, Rissing et al. demonstrated that 28% of IPNs progressed into overt lung metastases, with IPNs > 5 mm in size associated with decreased disease-free survival at 3 years [5]. Nakamura et al. analysed the factors associated with an increased likelihood of nodules being malignant rather than benign, finding that larger nodules, *n* > 1, bilateral distribution and first detection during follow up rather than at screening were more likely to prove malignant [3].

Differentiation between benign and metastatic lung nodules is of high clinical importance, as it helps to guide treatment decisions which have proven impacts on survival; Billingsley et al. demonstrated the complete resection of metastatic disease as being the most important prognostic factor in STS patients with pulmonary metastases [4]. Not only is this important for clinical decision making, but the detection of IPNs can add significant stress to patients who may not actually have metastatic disease. Whilst new technologies such as positron emission tomography (PET) show promise in the detection of metastatic disease [9], a study by Fortes et al. (including sarcoma patients) reported a 30% false negative rate in the detection of metastatic pulmonary nodules [10]. As such, there is an unmet clinical need both for the detection and interpretation of pulmonary nodules in patients with high-grade STS.

Whilst IPN rates have previously been reported in STS, we aimed to perform the largest, most in-depth analysis to date of IPNs, their progression and detection, as well as looking at their effect on patient overall survival (OS), focusing specifically on high-grade STS.

## 2. Materials and Methods

A retrospective cohort study of consecutive patients treated for high-grade soft tissue sarcoma in the North of England Bone and Soft Tissue Tumour Service between 1 January 2010 and 1 May 2020 was performed, following identification from a prospectively maintained database. This study was registered with the local institutional review board (number 13952). Low-grade sarcomas, as well as tumours arising from visceral, retroperitoneal and intracranial locations, were excluded from the study. Patients with no available staging scans for review were also excluded. All grading and classification of tumour subtypes were conducted by expert sarcoma pathologists, according to the WHO classification of bone and soft tissue tumours [11]. Tumours were considered to be high-grade if they were scored as being grade 2 or 3 using the French Fédération Nationale des Centres de Lutte Contre le Cancer (FNCLCC) grading system [12]. As the FNCLCC grading system does not apply to all sarcoma subtypes, sarcomas reported as being morphologically high-grade in the pathology report were also included.

Patients underwent staging as part of their initial work up; this was usually performed at our centre using a Scanner Somatom Definition AS by Siemens, Erlangen, Germany—3 mm slices, although this was occasionally performed at local hospitals due to the logistics of travel. The scans and reports were reviewed by the lead author (MJB), whilst blinded to the outcomes at this point. Nodules were classified as metastatic, benign or indeterminate, with indeterminate pulmonary nodules defined as non-calcified nodules < 10 mm in maximum diameter [5]. Follow up imaging was reviewed to determine whether these nodules remained unchanged or progressed, as well as to monitor for the development of a new disease. Survival data were also collected by reviewing patients’ clinical notes and collecting information including age, gender, tumour location, histological subtype, grade, size and depth relative to the fascia.

Differences between groups were compared using independent T-tests and Fisher’s exact test, accordingly, using SPSS statistics (Version 28.0, IBM Corp, Armonk, NY, USA). The time taken to develop lung metastases in different groups was analysed using Kaplan–Meier plots and log rank tests to calculate *p* values. The influence of other risk factors on the progression of IPNs and the development of metastases was analysed as part of both univariate and multivariate models using the Cox proportional hazards model. Survival analysis was conducted using R statistics (version 4.2.1, R Foundation for Statistical Computing, Vienna, Austria).

## 3. Results

A total of 389 patients were identified as being suitable for the study and their basic clinical information is displayed in Table 1. Patients had a mean age of 61.9 years (range 2–97), with a male predominance of 62.4% of the cohort. A wide range of histological subtypes were included, with undifferentiated pleomorphic sarcoma (UPS) (25.7%) and myxofibrosarcoma (25.7%) being the most frequent subtypes. Only high-grade sarcomas were included, with 71.5% being grade 3, and the remaining 28.5% being grade 2.

After reviewing the staging CT scans and reports, 222 (57.1%) patients had no evidence of lung metastases or IPNs, 134 (34.4%) had IPNs and 33 (8.5%) had synchronous lung metastases (Figure 1). Of the patients with no IPNs or lung metastases upon CT staging, 62 (27.9%) went on to develop metastases at a later date, with a median time to development of lung metastases of 448 days (range 87–1998). A greater percentage of patients with IPNs at diagnosis went on to develop lung metastases, with 48 (35.8%) developing metastatic disease after a median of 249 days (range 13–1676). Patients with IPNs at diagnosis appeared to be at a higher risk of developing lung metastases than patients without, although this did not reach significance (Figure 2A, KM *p* = 0.14, HR 1.33, HR *p* = 0.14). When patients with grade 2 primaries were excluded, this became significant (Figure 2B, KM *p* = 0.019, HR 1.62, HR *p* = 0.021).

Of those patients with IPNs from the initial CT staging developing metastases in the future, 27 (56.3%) had progression of these IPNs, whereas the remaining 11 (43.7%) had new lesions, suggesting that only 20.1% of IPNs progressed. Of those with grade 3 primaries, progression occurred in 27.2%. The progression of known IPNs occurred sooner than the development of new lesions, with a median time to progression of 143 days (range 13–557) compared to 409 days (range 96–1676) (*p* < 0.001). Table 2 displays the distribution of risk factors amongst IPNs which did and did not progress; the progression group contained a significantly higher proportion of multiple IPNs (*p* = 0.010), IPNs ≥ 5 mm in diameter (*p* = 0.008), bilateral IPNs (*p =* 0.029), primaries ≥ 5 cm (*p* = 0.010), grade 3 primaries (*p* = 0.002) and primaries arising deep to the fascia (*p* = 0.032). When the above risk factors for IPN progression were analysed using a Cox regression model, all demonstrated a significant increase in the risk of progression to lung metastases (Table 3). When analysed at the multivariate level, IPNs ≥ 5 mm in diameter at diagnosis and grade 3 primaries retained significance (HR = 2.37, *p* = 0.03 and HR = 6.07, *p* = 0.015, respectively). Appendix A shows the clinical details of all of the patients with IPNs that progressed into metastases; only two patients had a grade 2 sarcoma, with the rest having grade 3 primaries, whilst only three patients had a primary sarcoma <5 cm. The median age of patients whose IPNs progressed was 68 years (range 11–87) and there was no significant difference in the average age between the group of patients whose IPNs progressed and those whose remained stable (*p* = 0.531). Of these 27 patients, the progression of IPNs was detected on interval CT scans in 9 patients, and six were detected upon routine surveillance chest X-ray (CXR) prior to confirmation with CT scanning of the chest; the remainder were detected on scans carried out for patients who were acutely unwell or during restaging following the detection of metastatic disease elsewhere.

Figure 3A displays the OS of the three patient groups, demonstrating a significant difference in the OS between the three groups (*p* < 0.001). When those presenting with frank metastases were removed, a trend towards poorer OS (Figure 3B) was seen in those with IPNs upon staging CT, although this did not reach significance (*p* = 0.19, HR = 1.23, HR *p* = 0.190). This remained insignificant at the multivariate level when analysed with known prognostic risk factors of tumour size (<5 cm or ≥5 cm), depth relative to the fascia and grade (Table 4). When patients with grade 2 primaries were excluded, worse OS was seen in patients presenting with IPNs at diagnosis (*p* = 0.016, HR 1.50, HR *p* = 0.017) (Figure 3C). All but one patient with IPNs that progressed are now deceased, with a median OS of 248 days (range 23–840).

The percentage of patients developing lung metastases varied between histological subtypes; during follow up, 53.6% of synovial sarcoma patients, 46% of UPS patients, 40.5% of leiomyosarcoma patients, 33.3% of liposarcoma patients, 28.9% of angiosarcoma patients and 20% of myxofibrosarcoma patients developed lung metastases. The cohort included 100 UPSs and 100 myxofibrosarcomas; as such, these were analysed independently as a subanalysis. Of the 100 patients with UPSs, 53 (53%) had no lung metastasis, 35 (35%) had IPNs and 12 (12%) had synchronous lung metastases. Of those with no lung metastases or IPNs at diagnosis, 18 (33.9%) developed lung metastases at a later date. Of those presenting with IPNs, 12 (34.2%) progressed into lung metastases, 4 (11.4%) developed new lung metastases and 18 (51.4%) remained stable, with no metastases developing elsewhere in the lungs. The presence of IPNs did not increase the likelihood of developing lung metastases (*p* = 0.45, Figure 4A), nor did it confer worse OS (*p* = 0.64, Figure 4B). Of the 100 patients included with myxofibrosarcoma, 66 (66%) presented with no lung metastases or IPNs, 32 (32%) presented with IPNs and 2 (2%) presented with lung metastases. Only 12 (18.2%) of the patients presenting without metastases or IPNs went on to develop metastases. Of those presenting with IPNs, 0 progressed and 6 (18.8%) patients developed lung metastases in different areas of the lung. There was no difference in the likelihood of developing lung metastases between those presenting with IPNs and those presenting with no signs of IPNs or lung metastases (*p* = 0.68, Figure 4C), and no difference in overall survival (*p* = 0.46, Figure 4D).

## 4. Discussion

This study provides the largest and most in-depth analysis of IPNs in patients presenting with high-grade STS and provides new data on an area which remains poorly understood. We included 389 patients and demonstrated that the presence of IPNs is associated with an increased risk of developing lung metastases and poorer OS in patients with grade 3 STS, but not grade 2 STS. In this cohort, 34.4% of patients had IPNs at diagnosis, and of which, 20.1% progressed after a median time of 143 days.

This study builds on work conducted by Saifuddin et al. [6], with similar inclusion criteria, but with a longer period of follow up and a significantly larger sample, allowing for greater study of the progression of IPNs, rather than frequency at staging, and focusing solely on high-grade STS. We had similar issues as these authors in terms of inconsistency in follow up scans, meaning the rate of detection of progression may be lower than the true value; the length of follow up here reduces the chances of this however. Our results demonstrated that grade 3 STS patients presenting with IPNs have significantly poorer survival than patients presenting with no metastatic disease, although this was not observed in patients with grade 2 primaries. Previous studies in other types of sarcoma have produced various results; Tsoi et al. demonstrated worse OS in patients with IPNs in osteosarcoma [13], whilst Ghosh et al. found no difference [14]. Tsoi et al. also studied the relevance of IPNs in Ewing sarcoma, finding no difference in OS [15]. It is important to note that Tsoi et al.’s sample size for their osteosarcoma study was significantly larger than the other two studies mentioned, allowing for much greater power to detect a difference.

This study has once again reinforced the clinical conundrum raised by IPNs and highlights an unmet clinical need. This is particularly important given that pulmonary nodules have been shown to have a higher risk of being malignant in sarcoma patients [16]. We analysed the factors associated with the progression of IPNs to frank metastatic disease; this occurred in 27 out of 134 patients (20.1%) with IPNs at diagnosis. The results demonstrated that IPNs ≥ 5 mm, multiple IPNs and a bilateral IPN distribution were more likely to be indicators of metastatic disease, as shown previously [3,6,16]. Interestingly, the two factors with the highest HR for progression actually related to the primary rather than the IPNs, with a primary size of ≥5 cm and a grade 3 primary having HRs of 4.55 and 6.79, respectively. Of the 27 patients with IPNs that progressed, only 3 had a primary <5 cm and only 2 had a grade 2 primary. As suggested previously by Mayo et al. [17], this suggests that it is important to consider the characteristics of the primary as well as the IPN when considering the risk of progression; this is logical given that larger, higher-grade sarcomas are associated with higher rates of metastasis in general [18]. As such, even single IPNs < 5 mm warrant close surveillance in patients with large grade 3 primaries. The small number of IPNs progressing prohibited examining prognostic factors separately in different subtypes. The results demonstrated variation between the percentage of patients developing lung metastases during follow up however, with synovial sarcoma and UPS having particularly high rates; it would be logical that IPNs in these patients also had a higher risk of progression.

There are currently no established guidelines to guide the follow up of IPNs on CT staging scans in STS patients. Often, an interval scan at 3, 6 or 12 months is recommended by radiologists to look for changes in the nodules. Our results would suggest that this is insufficient however; 5 of the 27 IPNs that progressed did so after more than 12 months. Given that the median time to progression was 143 days, it seems reasonable that the initial interval scan should be performed within 6 months and be repeated after 12. Interestingly, progression was detected via surveillance CXR in 6 out of 27 patients who progressed. This highlights the importance of regular chest surveillance in STS patients, something which the SAFETY trial is currently investigating [19]. Gamboa et al. previously investigated surveillance methods in high-grade STS patients, comparing CT surveillance to CXR surveillance, finding no improvement in detection and intervention rates in the CT arm [20].

It is important that imaging technologies advance in order to aid clinical decision making in the management of these complex patients. The early identification of malignant nodules followed by rapid intervention with metastasectomy or stereotactic radiotherapy may increase the cure rate in these patients. Relatively new technologies such as FDG PET are still insufficient, with a false negative rate of 30% [10]. One possible solution to this is the development of targeted agents specific to surface proteins expressed on sarcoma cells, such as MT1-MMP [21,22,23,24]. Pringle et al. recently published preclinical data of a targeted MT1-MMP antibody labelled with both IRDye800 and Zr-DFO [25]. Zr-DFO emits Cerenkov luminescence and can be imaged via PET scans [26], offering the potential for more accurate pre-operative local imaging and theoretically a differentiation between IPNs and metastatic sarcoma deposits in the lungs. Furthermore, IRDye800 fluoresces in a similar spectrum to indocyanine green, which is currently under investigation for its utility in fluorescence-guided sarcoma surgery [27,28], meaning it could also be used as a targeted fluorescent dye for intra-operative guidance using current camera systems. Given that surgical resection is the current cornerstone of curative STS management, this dual purpose is particularly enticing. In combination with the ever-evolving fields of machine learning and artificial intelligence, which have already been suggested to be equal in efficacy at identifying pulmonary nodules to consultant radiologists, huge progress could be made in the distinction of malignant and benign pulmonary nodules over the coming years [29,30]. Ideally, this will remove the concept of IPNs, being able to accurately distinguish between metastatic disease and benign nodules.

## 5. Conclusions

In conclusion, this study demonstrates that patients with grade 3 STS presenting with IPNs have significantly worse survival than those without. It also highlights the importance of the consideration of factors related to the primary tumour itself when evaluating the risk of IPN progression in patients with high-grade STS; IPNs in patients with larger, higher-grade primaries arising deep in the fascia are associated with an increased risk of progression, as are IPNs ≥ 5 mm in diameter, multiple IPNs and a bilateral distribution. In order to monitor for progression, we recommend that IPNs are followed up with CT scans at 6 and 12 months. Further study in a larger cohort of people with high-grade STS is required, particularly to allow for the analysis of the role of subtype on risk of progression.

## Figures and Tables

**Figure 1 cancers-15-03531-f001:**
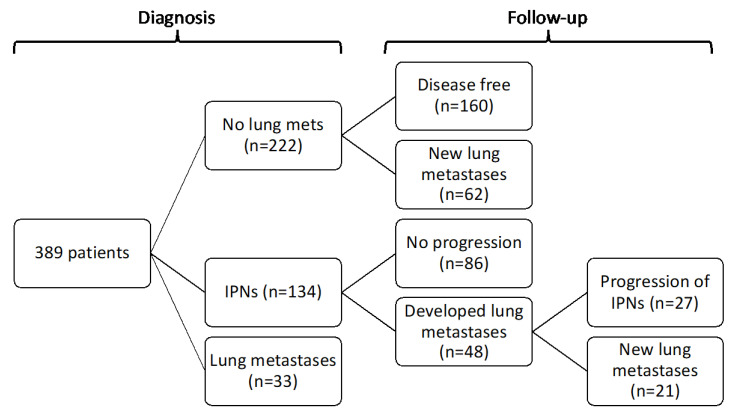
Flowchart depicting patients’ IPN statuses at diagnosis and their progression during follow up.

**Figure 2 cancers-15-03531-f002:**
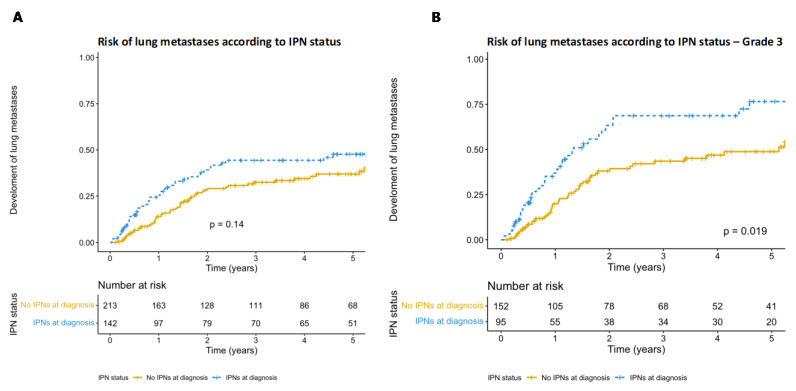
(**A**) Kaplan–Meier graph depicting the cumulative hazard of the development of lung metastases in patients with and without IPNs on the CT staging scan. Patients with IPNs at diagnosis appeared to be at higher risk of developing lung metastases, but this did not reach significance (*p* = 0.14, HR 1.33, HR *p* = 0.14). (**B**) Kaplan–Meier graph depicting the risk of lung metastases according to IPN status at diagnosis after patients with grade 2 primaries were excluded. Patients with IPNs at diagnosis are at a significantly higher risk of developing lung metastases (*p* = 0.019, HR 1.62, HR *p* = 0.021).

**Figure 3 cancers-15-03531-f003:**
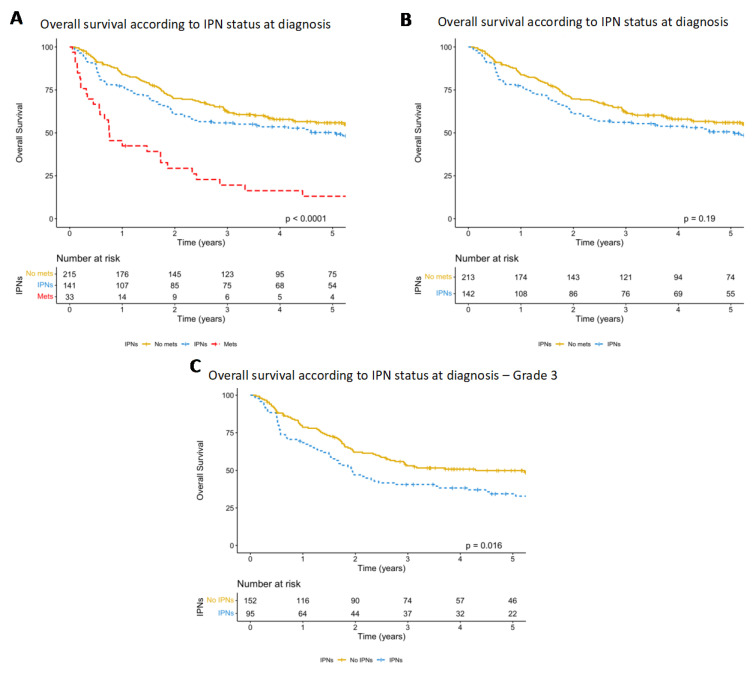
(**A**) Kaplan–Meier graph comparing survival in patients with no IPNs or lung metastases at diagnosis, IPNs and overt lung metastases, with a significant difference detected between groups (*p* < 0.001). (**B**) The same Kaplan–Meier graph with the lung metastases at presentation group removed, demonstrating a trend to decreased OS in patients with IPNs at diagnosis, although this did not reach significance (*p* = 0.19). (**C**) OS according to IPN status once patients with grade 2 primaries are excluded, demonstrating significantly decreased OS in patients with IPNs at diagnosis (*p* = 0.016).

**Figure 4 cancers-15-03531-f004:**
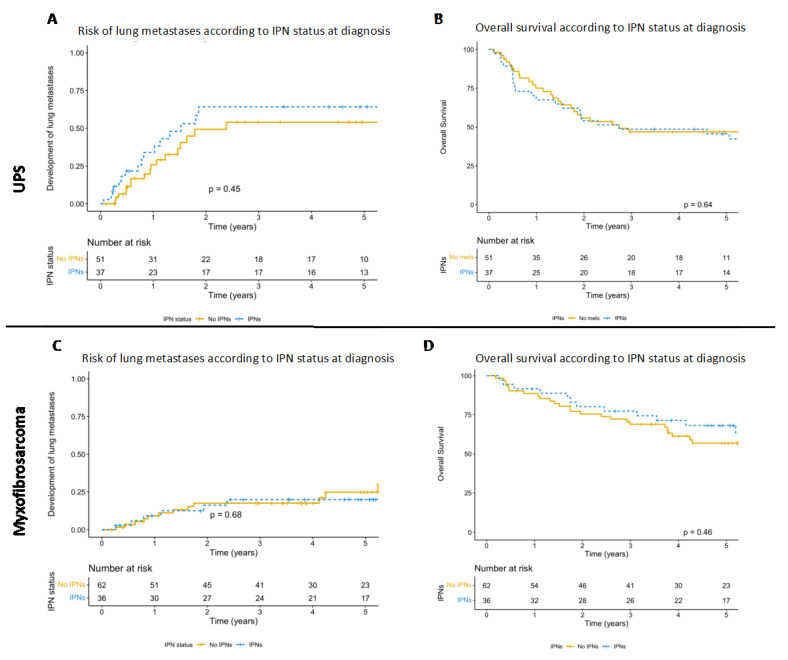
(**A**) Kaplan–Meier graph depicting the cumulative hazard of the development of lung metastases in UPS patients with and without IPNs on the CT staging scan. Patients with IPNs at diagnosis were not at increased risk of developing lung metastases (*p* = 0.45). (**B**) Kaplan–Meier graph comparing survival in UPS patients with no IPNs and IPNs detected upon CT staging, with no significant difference between groups seen (*p* = 0.64). (**C**) Kaplan–Meier graph depicting the cumulative hazard of the development of lung metastases in myxofibrosarcoma patients with and without IPNs on the CT staging scan. Patients with IPNs at diagnosis did not have an increased risk of developing lung metastases (*p* = 0.68). (**D**) Kaplan–Meier graph comparing survival in myxofibrosarcoma patients with no IPNs and IPNs detected upon CT staging, with no significant difference seen between groups (*p* = 0.46).

**Table 1 cancers-15-03531-t001:** Summary of demographic and basic clinical information.

Characteristic	
**Mean age, years (range)**	61.9 (2–97)
**Gender, number (%)**	
Male	243 (62.4%)
Female	146 (37.6%)
**Location, number (%)**	
Lower limb	223 (57.3%)
Upper limb	69 (17.7%)
Trunk	87 (22.4%)
Head and neck	10 (2.6%)
**Histological subtype, number (%)**	
Angiosarcoma	38 (9.8%)
Extra-skeletal Ewing sarcoma	9 (2.3%)
Leiomyosarcoma	42 (10.8%)
Liposarcoma	32 (8.2%)
MPNST	0 (2.6%)
Myxofibrosarcoma	100 (25.7%)
Rhabdomyosarcoma	19 (4.9%)
Synovial sarcoma	28 (7.2%)
Undifferentiated pleomorphic sarcoma	100 (25.7%)
Other	11 (3.8%)
**FNCLCC grade, number (%)**	
Grade 2	111 (28.5%)
Grade 3	278 (71.5%)
**Size**	
<5 cm	121 (31.1%)
≥5 cm	268 (68.9%)
**Depth relative to fascia**	
Superficial	179 (46.0%)
Deep	210 (54.0%)

**Table 2 cancers-15-03531-t002:** Distribution of risk factors amongst patients with IPNs that did and did not progress, *p* value calculated using Fisher’s exact test. * indicates significance.

		Clinical Outcome of IPNs (*n* = 134)		
		Stable (*n* = 107)	Progressed (*n* = 27)	*p* Value
**IPN number**	1	66	9	**0.010 ***
	>1	41	18	
**IPN size ≥ 5 mm**	1	74	11	**0.008 ***
	>1	33	16	
**Bilateral IPNs**	No	83	15	**0.029 ***
	Yes	24	12	
**Primary size ≥ 5 cm**	No	40	3	**0.010 ***
	Yes	67	24	
**Primary grade**	2	40	2	**0.002 ***
	3	67	25	
**Primary depth**	Superficial	53	7	**0.032 ***
	Deep	54	20	

**Table 3 cancers-15-03531-t003:** Analysis of IPN and tumour characteristics on the progression of IPNs to metastatic disease, calculated using the Cox regression model. * indicates significance.

	Progression of IPNs			
	Univariate		Multivariate	
	HR	*p* Value	HR	*p* Value
**IPN number ≥ 1**	2.76	**0.013 ***	2.29	0.087
**IPN size ≥ 5 mm**	2.96	**0.006 ***	2.37	**0.030 ***
**Bilateral IPNs**	2.43	**0.022 ***	1.66	0.282
**Primary size ≥ 5 cm**	4.55	**0.014 ***	2.58	0.160
**Primary grade**	6.79	**0.009 ***	6.07	**0.015 ***
**Primary depth**	2.46	**0.041 ***	1.71	0.280

**Table 4 cancers-15-03531-t004:** Analysis of tumour characteristics and IPN status (IPNs or no IPNs/lung metastases) at diagnosis on the progression of IPNs to metastatic disease, calculated using the Cox regression model. * indicates significance.

	Overall Survival			
	Univariate		Multivariate	
	HR	*p* Value	HR	*p* Value
**Primary size ≥ 5 cm**	3.60	**<0.001 ***	3.03	**<0.001 ***
**Primary grade**	2.74	**<0.001 ***	2.25	**<0.001 ***
**Primary depth**	1.48	**0.011 ***	1.01	0.947
**IPN status**	1.23	0.190	1.31	0.080

## Data Availability

Further data are available upon request from the corresponding author.

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
