# Peer review of "What Is the Significance of Indeterminate Pulmonary Nodules in High-Grade Soft Tissue Sarcomas? A Retrospective Cohort Study"

_cancers, 2023, doi:10.3390/cancers15133531_

Round 1

Reviewer 1 Report

It is of little clinical value to discuss the significance of indeterminate pulmonary nodules (IPNs) by collecting sarcomas with various clinical behaviors.

Soft tissue sarcomas differ in age of onset, indications for chemotherapy and drug sensitivity depending on histological type. In addition, the risk of recurrence and metastasis and prognosis vary greatly depending on whether R0 resection is achieved, but these points have not been examined. For example, in the case of IPNs, metastasis is strongly suspected if the patient is young and has synovial sarcoma, extraskeletal Ewing sarcoma, or rhabdomyosarcoma, while metastasis is considered unlikely if the patient is elderly and has undifferentiated pleomorphic sarcoma (UPS) or myxofibrosarcoma (MFS). In addition, IPNs often disappear with intensive chemotherapy in young patients with round cell sarcoma. This means that histological type and age are important factors in determining whether IPNs are metastatic. Therefore, when discussing the significance of IPNs, each histological type should be evaluated separately. The authors refer to UPS and MFS in Fig. 4, and this part is thought-provoking. This is obvious when compared to the article by Tsoi cited in the discussion, which is limited to osteosarcoma. In that article, it is clearly stated that patients with IPNs showed worse overall survival.

In the case of IPNs, the risk of metastasis due to enlargement of the nodule should be clarified by each histological type.

Author Response

Thank you for your helpful feedback. Since the previous draft, we have expanded our cohort to 389 patients, making this the largest IPN study in STS to date that we are aware of. Your comments r.e. grouping multiple subtypes are valid and we have made attempts to address this, including the aforementioned cohort expansion. Unfortunately, there is insufficient numbers of patients with each subtype to analyse risk factors for progression independently, given that only the patients presenting with IPNs that then progress are included in that analysis (27 out of 389 in this cohort). We feel the factors identified as increasing the risk of progression of IPNs are likely to be consistent across high grade soft tissue sarcomas, with the varying subtypes representing an additional risk factor, so believe this is still of great value. We have therefore included the percentage of patients in each common subtype included that develop lung metastases during follow up as a marker of this, demonstrating which subtypes are more likely to be at higher risk of progression. In order to achieve sufficient numbers to include subtype as a risk factor in our analysis, we could group the subtypes with high and low rates of lung metastasis together. However, given the low numbers of some of these subtypes, we feel this could be inaccurate.

With regards to the inclusion of resection margins as a risk factor, we have not included this as we are looking at the progression of IPNs on CT staging scans; these were conducted prior to the resection so this is not directly related to whether this is metastatic, although of course it will increase their risk of developing other lung metastases. Age was considered, but there is no difference in the age of the patients between the groups of patients with stable IPNs and IPNs that progressed, hence it was not included in the analysis, although we have added this information to the results. Chemotherapy is used very sparingly in patients with localised soft tissue sarcoma in the UK; only 7 of the 134 patients with IPNs received chemotherapy. It would only be used for extra skeletal Ewing sarcomas as well as young patients with rhabdomyosarcomas and synovial sarcomas; collectively this made up <10% of our population so will not have a significant influence here.

We hope you feel that we have adequately addressed your feedback and look forward to hearing from you.

Reviewer 2 Report

The authors performed a retrospective cohort analysis of 200 patients with high grade sarcoma and IPNs. The authors evaluate the presence of IPNs with the progression to and/or the new development of lung metastases. In addition, IPNs were evaluated with respect to a variety pf clinical outcomes of interest, including survival.

The manuscript is well written and appropriately referenced. There are minor grammatically errors noted. The tables and figures referenced and well-designed and easy to understand.

The results are consistent with some prior published data, but this study does provide some new information. Appropriate limitations of the study are mentioned.

Comments:

1. Authors mention FNCC grade in text and Troajni grade in the Table. While they are referring to the same thing, I would suggest picking one nomenclature/designation and be consistent throughout the manuscript. (FNCC favored).

2. It is unclear if the determination of progression to lung metastases was based on radiographic growth alone, formal biopsy/confirmation of metastatic disease, or both. What percentage of patients with progressing IPNs underwent biopsy confirmation?

Minor changes:

1. Abstract: Simple summary, "effected" should be "affected"

Author Response

Thank you for your very helpful comments. Following comments from other reviewers, the case series has been expanded to 389 patients and the results updated accordingly, strengthening the study. We have corrected Trojani to FNCLCC as suggested. With regards to histological confirmation. This was usually based on radiologic enlargement and patients weren’t routinely biopsied. They were only biopsied if there was uncertainty or they had a second primary malignancy and this may have changed management. 2 of the patients whose IPNs progressed underwent wedge resections of the nodules and as such had the histological confirmation in that way. A further 3 had cytology performed after malignant pleural effusions were drained, showing the presence of metastatic disease, although of course this isn’t examining the specific nodule. The remaining 22/27 patients with progression were based on radiologic findings alone. We hope that you feel we have sufficiently addressed your comments and look forward to hearing from you.

Reviewer 3 Report

The authors present a review of patients with high- and intermediate-grade soft-tissue sarcomas with indeterminate pulmonary nodules from a database from 2010-2016.  Many of their conclusions fail to reach statistical significance due to limited sample size.  Since the median time to development of pulmonary metastases was about 1 year, updating the database to at least 2020 (or 2019 to decrease the confounding factor of COVID) would markedly strengthen the analysis.  Suggesting that there was no effect on survival because the p value was 0.11 is absurd, and since many readers may not get past the abstract, that's what they will come away with.

Minor improvements would help.  Some but not all examples follow.

How can patients have a scan of their chest?  Certainly they do not share one.

What is a diagnosis CT staging scan?  Better, the staging CT scan at diagnosis.

Author Response

Thank you for your helpful comments. As suggested, we have expanded the cohort to May 2020, increasing the sample size to 389; we feel this has greatly increased the quality of the study and allowed us to reach significance with regards to the predictors of progression and makes this the largest IPN study in STS patients to date that we are aware of. With regards to your point about suggesting in the abstract that there was no effect on survival being absurd we are a little confused; we do not feel that ‘trended towards decreased overall survival (p=0.11)’ in the results section of the abstract or ‘the effect on survival is unclear’ in the conclusion of the abstract section ever suggested this. We have made this is clear when editing the abstract with the updated results, however, whilst the large increase in cohort size allows more confidence in our findings. In addition, we feel that scoring of the English of the paper as incomprehensible is unjust; English is the first language of all authors and the other reviews have raised no such problems. We have proofread and edited the draft however and hope that you feel it has been sufficiently improved. We hope that you feel we have sufficiently addressed your comments and look forward to hearing from you.

Round 2

Reviewer 1 Report

Further analysis about the risk of metastasis in each histological type is expected. 

Author Response

Thank you for your further review of our manuscript. We have made a further addition to the manuscript based on the recommendation based on the other reviewer’s feedback, which shows that the presence of IPNs at diagnosis in patients with grade 3 STS (grade 2 excluded) is associated with poorer overall survival. With regards to your comment r.e. further analysis of each histological subtype, this has not been performed for the same reasons as described in our previous response. Unfortunately, despite the very large cohort, there are insufficient numbers present in each subtype to allow meaningful sub analysis of the IPNs that progress (only 27 patients in the whole cohort had IPNs that progressed). The factors associated with increased risk of progression are likely to be applicable to all subtypes and can be applied in context of the overall metastatic rates for each subtype which have been included in the results section. If there is a specific analysis you would like us to perform, we would be more than happy to consider this but, for the reasons described above, we cannot think of anything that will add value here.

Despite this, this manuscript contains important findings with regards to survival, risk of progression and surveillance needs which we feel are important to share with the sarcoma community. As such, we hope you now feel that it is suitable for publication in this journal.

Reviewer 3 Report

This is a markedly improved manuscript due to the additional patients included in the analysis.  The authors might consider revising their inclusion of patients with grade 2 sarcomas as high grade in regard to development of pulmonary metastases as one of their conclusions.  Their data indicate that for  patients in whom pulmonary metastases are a significant risk, the presence of indeterminate pulmonary nodules increases that risk.

I am not sure whether analyses is a verb in British English; it is not in American English.

Patients can undergo CT's of THE chest, but no patients can undergo CT's of their chest since they have chests but cannot share a single chest.

On page 4, commenting on the data on Figure 2, the authors state, "Figure 2, KM p=0.14, HR 1.33, HR p=0.014."  Is the second p value a typo?

Author Response

Thank you for your input. Your point about the inclusion of grade 2 primaries is very interesting. We have kept them included as it helps us to highlight the important fact that grade 3 primaries are at a much higher risk of progression, which is important to be published. We have however independently analysed survival in grade 3 primaries, and this has shown that IPNs at diagnosis are associated with decreased OS in these patients; this has been added to the paper accordingly.

Grammatical points:

  • Analyses is a verb in British English – it is the 3rd person present of the verb to analyse.
  • The use of the word ‘their’ is correct – ‘their’ simply indicates ownership and is therefore the correct word to use here; it does not mean that there is one single chest shared between the patient, rather that the scan is of their individual chest. Using ‘the’ instead would be incorrect as it would refer to any chest, not specifically the patient’s chest.
  • You are correct, this is a typo and has been corrected.

We hope that you now consider our manuscript suitable for publication.